# FEUSNet: Fourier Embedded U-Shaped Network for Image Denoising

**DOI:** 10.3390/e25101418

**Published:** 2023-10-05

**Authors:** Xi Li, Jingwei Han, Quan Yuan, Yaozong Zhang, Zhongtao Fu, Miao Zou, Zhenghua Huang

**Affiliations:** 1School of Electrical and Information Engineering, Wuhan Institute of Technology, Wuhan 430205, China; lixi@wit.edu.cn (X.L.); zoumiao@stu.wit.edu.cn (M.Z.); 2College of Information and Artificial Intelligence, Nanchang Institute of Science and Technology, Nanchang 330108, China; zhangyaozong@wit.edu.cn (Y.Z.); zhongtao.fu@wit.edu.cn (Z.F.); zhhuang@wit.edu.cn (Z.H.)

**Keywords:** deep convolution neural network, end-to-end denoising network mechanism, Fourier coefficients

## Abstract

Deep convolution neural networks have proven their powerful ability in comparing many tasks of computer vision due to their strong data learning capacity. In this paper, we propose a novel end-to-end denoising network, termed Fourier embedded U-shaped network (FEUSNet). By analyzing the amplitude spectrum and phase spectrum of Fourier coefficients, we find that low-frequency features of an image are in the former while noise features are in the latter. To make full use of this characteristic, Fourier features are learned and are concatenated as a prior module that is embedded into a U-shaped network to reduce noise while preserving multi-scale fine details. In the experiments, we first present ablation studies on the Fourier coefficients’ learning networks and loss function. Then, we compare the proposed FEUSNet with the state-of-the-art denoising methods in quantization and qualification. The experimental results show that our FEUSNet performs well in noise suppression and preserves multi-scale enjoyable structures, even outperforming advanced denoising approaches.

## 1. Introduction

Image denoising [1], a fundamental and important issue in low-level vision and image processing, aims at removing or eliminating external noise as much as possible while preserving clear details in the original image. Its essence lies in the process of reducing noise in the image, restoring and reconstructing the original clear image. While image restoration [2] is a long-standing problem, in the general image restoration problem a damaged image Y can be expressed as follows:(1)$$\begin{array}{c} Y=T\left(X\right)+n \end{array}$$
where X represents a clear image, T(•) represents a degenerate function, and n represents additive noise. Generally referring to additive noise, image denoising is a common restoration technique.

In the early days, the most representative methods of traditional image denoising were block-matching and 3D filtering [3] (BM3D) and non-local means [4] (NLM), among others [5]. However, in recent years, deep learning-based image denoising methods have surpassed traditional image denoising methods [6] in terms of inference time and denoising performance. Early deep learning image denoising methods used reinforcement learning techniques [7], such as Q-learning [8] and other training recursive neural networks. However, reinforcement learning-based methods require a large amount of computation and have low search efficiency. Currently, deep learning denoising methods combine skip connections [9], attention mechanisms [10], multiscale feature fusion [11], and the introduction of residual blocks [12] to improve the network feature expression capabilities. Current methods for image denoising can be roughly divided into two categories: image denoising based on traditional methods and image denoising based on deep learning. For example, bilateral filters [13], Gaussian filters [14], and median filtering [15] are traditional image denoising methods. Discrete cosine transform [16], wavelet transform [17], and other methods are also used to modify the transform coefficients [18], and the average neighborhood [19] values are utilized to calculate the local similarity [20]. These methods are based on image denoising and attempt to preserve more edge details using smooth image features. However, the images processed with these methods often become blurry, and the edge details of the original image are not clearly retained, resulting in a poor overall effect.

With the development of deep learning, neural networks have overcome the drawbacks of traditional denoising methods. Most deep learning-based methods are external prior methods [21]. In 2017, Zhang et al. [22] proposed a convolutional neural network (CNN) called DnCNN, which utilizes residual learning and batch normalization to achieve network denoising. In 2018, Zhang et al. [23] proposed a faster and more flexible denoising convolutional neural network called FFDNet, which can remove more complex noise. In 2019, Guo et al. [24] proposed a real image-blind denoising network called CBDNet. They trained the network using synthetic and real-world noise images, dividing it into two subnetworks: nonblind denoising and noise estimation, which improved the generalization ability of deep CNN denoisers [25]. Presently, deep learning-based denoisers [26,27,28,29] have achieved good results, but most of these networks execute CNNs in the spatial domain. In recent years, transformer models have been successful in natural language processing (NLP). Visual transformers [30] have been widely used in image restoration [31] tasks owing to their strong global modeling ability. In 2022, Fan et al. [32] proposed the SUNet network, which combined a Swin transformer [33] and UNet into a denoising model and demonstrated impressive performance in image denoising tasks. Although these methods [34] have outstanding image denoising capabilities, they overlook the inherent priors of noisy images, making them prone to overfitting in synthetic datasets.

To date, some researchers have applied the Fourier transform to other low-level visual tasks, such as image deblurring [35] and image deraining [36]. We introduced a Fourier transform into the field of image denoising and learned the frequency domain features of the images. We equipped our Fourier transform residual blocks with a simple three-layer UNet [37]. A more complex network structure may result in greater performance improvement. Compared to the structural information, additive Gaussian white noise has a higher frequency. According to experimental results, we find that such a conclusion is reasonable. In view of this, we tried to plug the Fourier transform into a U-shaped network for a noise removal model construction, and the experimental results demonstrate that the proposed method can achieve promising results. Meanwhile, we also present their time cost. Compared with current mainstream deep learning-based image denoising methods, our network can achieve better performance, reflecting the superiority of our Fourier prior in image denoising tasks.

The main contributions of this study can be summarized as follows:We propose a Fourier prior for image denoising that includes the physical characteristics of noisy images in both the spatial and frequency domains;We designed and implemented a simple and effective residual block based on the Fourier transform that processes the amplitude and phase spectra of noisy images in parallel within Res FFT blocks and learns the frequency domain features of noisy images.

## 2. Fourier Embedded U-Shaped Network

We first introduce the Fourier prior in Section 2.1, where we conjecture and prove the well-known characteristics [38,39] of amplitude and phase spectra in noisy images. In Section 2.2, we present our proposed Res FFT blocks. In Section 2.3, we introduce our network structure. The loss function used for training is described in Section 2.4.

### 2.1. Fourier Prior

Mathematically, the Fourier transform refers to the ability to represent a function that satisfies certain conditions as a linear combination of a series of sine or cosine functions. When the Fourier transform is applied to image operations from a physical perspective, it transforms an image from the spatial domain to the frequency domain, whereas its inverse transformation transforms the image from the frequency domain to the spatial domain. Given image $f\left(x,y\right)\in R^{H\times W\times 1}$, the Fourier transform is represented as
(2)$$\begin{array}{c} F\left(u,v\right)=\frac{1}{HW}\sum_{x=0}^{H-1}\sum_{y=0}^{W-1}f\left(x,y\right)e^{-j2\pi \left(\frac{ux}{H}+\frac{vy}{W}\right)} \end{array}$$
where u=0,1,2,…,H−1 and v=0,1,2,…,W−1. Similarly, given $F\left(u,v\right)$, $f\left(x,y\right)$ can be obtained through an inverse Fourier transform, which can be formulated as
(3)$$\begin{array}{c} f\left(x,y\right)=\sum_{x=0}^{H-1}\sum_{y=0}^{W-1}F\left(u,v\right)e^{j2\pi \left(\frac{ux}{H}+\frac{vy}{W}\right)} \end{array}$$
where x=0,1,2,…,H−1 and y=0,1,2,…,W−1. Given $f\left(x,y\right)$, the amplitude spectrum $\left|F\left(u,v\right)\right|$ and the phase spectrum $\varphi \left(u,v\right)$ can be obtained after the Fourier transform as follows: (4)$$\begin{array}{cc} \left|F\left(u,v\right)\right|= & \sqrt{{Re}^{2}\left(u,v\right)+{Im}^{2}\left(u,v\right)} \end{array}$$(5)$$\begin{array}{cc} \varphi \left(u,v\right) & =\arctan\left[\frac{Im(u,v)}{Re(u,v)}\right] \end{array}$$ Here, Re(u,v) and Im(u,v) represent the real and imaginary parts of $F\left(u,v\right)$, respectively.

By comparing the visualized images in Figure 1, we found that, in terms of visual perception, there was no significant difference in the real, (a) and (e), and imaginary, (b) and (f), parts of both the noisy image and the ground truth after the Fourier transform. However, there is a small difference in the amplitude spectra, (c) and (g), between the noisy image and the ground truth, and a significant difference in the phase spectra, (d) and (h), between the noisy image and the ground truth. Therefore, we infer that the noise features of the image may mostly exist in the phase spectrum of the image and that a small portion of the noise features may also exist in the amplitude spectrum of the image.

In Figure 2, we can observe from the results in (a) and (b) that the amplitude spectrum of the image represents the brightness of each pixel in the image. The center of the amplitude spectrum is the low-frequency region; the higher the brightness of the image is, the larger the corresponding amplitude spectrum value. That is, the amplitude spectrum stores the amplitude information of each pixel in the image, but the position information of the original pixel has been disrupted, and the original image cannot be reconstructed solely by the amplitude spectrum of the image. We can see from the results of (c) and (d) that the phase spectrum of the image records the position information of each pixel in the image. By observing the phase spectrum of the image for visualization operation in Figure 1d,h, the phase spectrum resembles a cluster of noise, but it is also particularly important for image reconstruction, and the original image cannot be reconstructed solely from the phase spectrum of the image. We then compare (e) and (f) in Figure 2. According to the rotation invariance of the Fourier transform, when the amplitude spectrum of the noise image is rotated 180° and the phase spectrum of the original noise image is reconstructed, it can be seen with the naked eye that the overall image does not rotate.

When the phase spectrum of the noise image is rotated 180° and the amplitude spectrum of the original noise image is reconstructed, it can be seen with the naked eye that the overall image is rotated 180°, thus verifying the conclusions of the above two points regarding the amplitude spectrum and phase spectrum of the image. To further prove our hypothesis, we found that (g) and (h) in Figure 2 compared the original noisy image with the labeled image, and (g) and (h) showed varying degrees of noise reduction visible to the naked eye. The noise level of (h) was significantly higher than that of (g), and (g) exhibited a decrease in image brightness. As mentioned previously, we can safely conclude that most noise features of the image exist in the phase spectrum of the image.

### 2.2. Res FFT Blocks

A widely used residual Fast Fourier Transform module based on ReLU is only utilized to concatenate the real and imaginary parts in the last dimension after the Fourier transform. However, it ignores the respective roles of the real and imaginary parts of the Fourier coefficients in the image, as shown in Figure 3a. We propose an improved Res FFT block, in which we preserve the identity mapping and normal spatial residual edges for auxiliary network training. To utilize the Fourier priors, we used dual channels in the channels of the Fourier transform to process the amplitude and phase spectra in parallel, known as RFAPB, where we used eight cascaded residual blocks in the ERB and DRB. The RFAPB structure is shown in Figure 3b.

We set $X\in R^{H\times W\times C}$ as the input feature graph, where *H*, *W*, and *C* are the height, width, and number of channels of the feature graph, respectively. The overall data flow processing of RFAPB is as follows: (1) Input feature map $X\in R^{H\times W\times C}$. (2) (i) Fourier transform flow: calculate the two-dimensional discrete Fourier transform of X to obtain $F\left(X\right)\in C^{H\times W\times C}$; take the real part R[F(X)] of the Fourier coefficient and the imaginary part I[F(X)] of the Fourier coefficient and calculate the amplitude spectrum A[F(X)] and the phase spectrum P[F(X)] based on the real and imaginary parts of the Fourier coefficient. Two stacked 1×1 convolution layers (convolution operator ⊙) and a ReLU activation function are used in the middle to process the amplitude spectrum A[F(X)] and the phase spectrum P[F(X)], respectively. The processing part of the amplitude spectrum is formulated as
(6)$$\begin{array}{c} f\left\{A\left[F\left(X\right)\right],C^{1},C^{2}\right\}=\mathrm{ReLU}(A\left[F\left(X\right)\right]⊙C^{1})⊙C^{2} \end{array}$$
where $f\left\{A\left[F\left(X\right)\right],C^{1},C^{2}\right\}\in C^{H\times W\times C}$. The processing part of the phase spectrum is formulated as
(7)$$\begin{array}{c} f\left\{P\left[F\left(X\right)\right],C^{1},C^{2}\right\}=\mathrm{ReLU}(P\left[F\left(X\right)\right]⊙C^{1})⊙C^{2} \end{array}$$
where $f\left\{P\left[F\left(X\right)\right],C^{1},C^{2}\right\}\in C^{H\times W\times C}$. The feature graph $Y_{fft}\in R^{H\times W\times C}$ is reconstructed according to the amplitude spectrum and phase spectrum, and the reconstructed feature graph $Y_{fft}$ is calculated by using the two-dimensional inverse discrete Fourier transform, which can be formulated as
(8)$$\begin{array}{c} Y_{fft}=F^{-1}\left\{A\left[F\left(X\right)\right],P\left[F\left(X\right)\right]\right\} \end{array}$$ (ii) Main branch feature flow: Input feature map X through two stacked 3 × 3 convolution layers (convolution operator ⊙). A ReLU activation function is used in the middle, which can be formulated as
(9)$$\begin{array}{c} Y_{main}=g\left\{X,C^{1},C^{2}\right\}=\mathrm{ReLU}(X⊙C^{1})⊙C^{2} \end{array}$$ (iii) Short-cut branching: output feature map $X\in R^{H\times W\times C}$. (3) Output feature map of improved residual modules $Y=Y_{fft}+Y_{main}+X$, $Y\in R^{H\times W\times C}$, $Y_{fft}\in R^{H\times W\times C}$, $Y_{main}\in R^{H\times W\times C}$, and $X\in R^{H\times W\times C}$.

### 2.3. U-Shaped Network

The encoder–decoder structure is widely used in image denoising networks. The encoder structure refers to the gradual conversion of the input image data into feature maps with smaller spatial dimensions and more image channels, followed by the gradual conversion and restoration of the feature maps to the input image size through the decoder. This network structure is a symmetrical CNN, the most typical of which is a UNet network structure. In the encoder and decoder stages, a conventional skip connection method is used to combine different levels of information, which is conducive to the propagation of gradients and convergence of the model. The network structure diagram is shown in Figure 4.

In Figure 4, we not only use global residual learning but also introduce residual blocks for encoding and decoding. Here, we simply used the Res FFT blocks in the UNet architecture. We reviewed the reconstruction process of applying residuals to deep learning for image denoising. This method can also be used to build deep networks. Simultaneously, using multilevel residuals to stack, we can expand the receptive field of the feature image, which can be used to extract more delicate features in the image. After the Fourier transform, noisy images can be used to separate the low- and high-frequency features of the image, which is beneficial for preserving low-frequency features and removing high-frequency noise features. The structure of the residual network also solves the problem of network degradation to a certain extent and provides a simple mapping of the original features in the forward propagation process, which helps the model converge. In Figure 5, we present the intermediate results at different stages. The entire network is divided into two layers (Encode and Decode), and we visualize all the high-dimensional feature maps for each intermediate filter.

### 2.4. Loss Function

Because the mean square error (MSE) is the average square of the difference between the predicted value $\hat{X}$ of the model and the ground truth $X_{label}$ of the sample for image denoising tasks, high-frequency texture information may be lost during the training process because of the MSE penalty, resulting in blurred and overly smooth vision. Therefore, we used the conventional mean absolute error (MAE) to balance image noise removal with the preservation of detailed features.
(10)$$\begin{array}{c} Loss=L_{1}+\lambda L_{phase} \end{array}$$
where λ can be set according to the empirical value and λ is set to 0 in our experiment; $L_{1}=\frac{1}{n}\sum_{i=1}^{n}\left|\hat{X}-X_{label}\right|$, and $L_{phase}=\frac{1}{n}\sum_{i=1}^{n}\left|P\left(\hat{X}\right)-P\left(X_{label}\right)\right|$.

## 3. Experiments

### 3.1. Datasets

#### 3.1.1. Training Set and Validation Set

We trained our model using the DIV2K [40] dataset, which contains 900 high-resolution color images and is currently one of the most commonly used datasets for image super-resolution. We divided the 900 images in this dataset into 800 and 100 high-resolution images at an 8:1 ratio (with an average resolution of approximately 1920 × 1080). For the training set, we randomly cropped each training image into 10 pieces with a size of 256 × 256 patches and randomly applied additive Gaussian white noise (AWGN) to each patch with noise levels of σ = 5∼50 and a noise level interval of 5. For the validation set, we randomly cropped each image into three sizes of 256 × 256 patches and added AWGN with three different noise levels, σ=10, σ=30, and σ=50, to each patch. Therefore, 16,000 patches of size 256 × 256 were used to train the image denoising task, and 1800 patches of size 256 × 256 were used to validate the image denoising task. The dataset comprised 17,800 images.

#### 3.1.2. Testing Set

CBSD68 [41] is a dataset used to evaluate the performance of image denoising algorithms and is part of the Berkeley segmentation dataset and benchmark. The dataset includes 12 buildings; 30 animals, such as cats, tigers, and horses; 11 people; four plants and animals; and 11 other outdoor scene images. Kodak24 [42] mainly provides outdoor scene images, mostly from the perspective of buildings, sky, and sea. The scene images of BSD68 [43] and CBSD68 are the same, but the former are the grayscale versions of the images in the CBSD68 dataset. There are nine outdoor scene images and three indoor scene images in Set12, including five characters, three animals, and other outdoor scene images such as houses and ships. The CBSD68 and Kodak24 datasets were used to test the color image denoising model, whereas the BSD68 and Set12 [22] datasets were used to test the grayscale image denoising model. We tested our model on these four commonly used datasets (CBSD68, Kodak24, BSD68, and Set12). To test the impact of different noise intensities on the network performance, we added AWGN with noise levels of 10, 30, and 50 to these datasets.

### 3.2. Experiment Setup

#### 3.2.1. Implementation Details

All the experiments were conducted on a server equipped with a third-generation intelligent Intel Xeon processor and an NVIDIA Tesla A100 40G. Our model trained a patch of size 256 × 256, and it took approximately 30 h to train the synthetic noise images on the DIV2K dataset. During the training process, it was primarily used to remove the AWGN. The Adam optimizer was used to optimize the network parameters. The hyperparameter batch size for training was 12, and the initial learning rate was set to $2\times 10^{-4}$. Using the cosine annealing learning strategy, the minimum learning rate decreased to $1\times 10^{-6}$, and a total of 200 epochs were trained. We used the default settings for the other hyperparameters of the Adam optimizer.

#### 3.2.2. Evaluation Metric

To quantitatively compare the advantages and disadvantages of the denoising performance, we used the peak signal-to-noise ratio [44] (PSNR) and structural similarity [45] (SSIM) for quantitative evaluation and analysis. In recent research, new methods [46,47,48,49] for image quality assessment have been proposed, which have potential implications for the performance evaluation of image denoising algorithms.

The PSNR is currently the most widely used method in the field of image denoising, and it mainly represents the difference in pixel values between images, which can be formulated as
(11)$$\begin{array}{c} \mathrm{PSNR}=10\log_{10}\left[{\frac{\left(2^{n}-1\right)}{MSE}}^{2}\right] \end{array}$$
where *n* is the number of bits per pixel, usually n=8, which means that the pixel’s gray scale is 256 (in dB); MSE represents the mean square error of the current image $\hat{f}\left(i,j\right)$ and the reference image f(i,j), which can be formulated as
(12)$$\begin{array}{c} \mathrm{MSE}=\frac{1}{MN}\sum_{i=1}^{M}\sum_{j=1}^{N}{\left[\hat{f}\left(i,j\right)-f\left(i,j\right)\right]}^{2} \end{array}$$
where *M* and *N* represent the height and width of the image.

SSIM measures the quality of images from three aspects: luminance, contrast, and structural information of the sample image. The calculation formulas for the luminance (*l*), contrast (*c*), and image structure information (*s*) are as follows: (13)$$\begin{array}{c} l(x,y)=\frac{2u_{x}u_{y}+c_{1}}{u_{x}^{2}+u_{y}^{2}+c_{1}} \end{array}$$(14)$$\begin{array}{c} c(x,y)=\frac{2\sigma_{x}\sigma_{y}+c_{2}}{\sigma_{x}^{2}+\sigma_{y}^{2}+c_{2}} \end{array}$$(15)$$\begin{array}{c} s(x,y)=\frac{\sigma_{xy}+c_{3}}{\sigma_{x}\sigma_{y}+c_{3}} \end{array}$$ Here, $c_{3}=\frac{c_{2}}{2}$, $u_{x}$ represents the mean value of the pixels in image *x*, $u_{y}$ represents the mean of the pixels in image *y*, $\sigma_{x}^{2}$ represents the variance of the pixels in image *x*, $\sigma_{y}^{2}$ represents the variance of the pixels in image *y*, and $\sigma_{xy}$ represents the covariance of image *x* and image *y*. The calculation formula for SSIM is as follows:
(16)$$\begin{array}{c} \mathrm{SSIM}\left(x,y\right)={\left[l\left(x,y\right)\right]}^{\alpha }{\left[c\left(x,y\right)\right]}^{\beta }{\left[s\left(x,y\right)\right]}^{\gamma } \end{array}$$
where, α, β, and γ represent the weights of the three dimensions and generally α=β=γ=1. The value of SSIM should not exceed 1, and the closer it is to 1, the better the denoising effect on the image.

### 3.3. Ablation Experiment

We propose two structures, as shown in Figure 3. The structure in Figure 3c adds a convolution layer of 1 × 1 before the Fourier transform relative to the structure shown in Figure 3b. To verify the effectiveness of the Fourier transform residual blocks, we ablated the use of two structures and trained them using different loss functions. The results in Table 1 present the denoising effect when the residual structures of different Fourier transforms are matched with different loss functions. The loss function is shown in Equation (Equation 10), which changes λ to conduct ablation experiments based on the empirical values. We separately set λ=0,0.05,0.2,0.5; $L_{1phase}\in \mathrm{MAE}$, and $L_{2phase}\in \mathrm{MSE}$. These data represent the evaluation of the denoising model on the validation set, where Gaussian white noise was added at noise levels of 10, 30, and 50. When calculating the PSNR and SSIM, the noisy images of the entire validation set were averaged. According to the experimental results in Table 1, using the structure in Figure 3b and the loss function of Equation (Equation 10), the best denoising effect is achieved at λ=0. When the same structure and the same loss function are used, the denoising effect will decrease to varying degrees with an increase in λ. Using the same Fourier transform residual structure and different loss functions, when λ is the same, the utilization of $L_{1phase}$ in the phase part can often achieve a better denoising effect.

In Figure 6, we show the visualization results of the color image denoising models in different ablation experiments and enlarge the details. Figure 6a shows the Fourier residual structure of Figure 3b with loss function $L_{1}$; Figure 6b shows the Fourier residual structure of Figure 3b with loss function $L_{1}+0.05L_{1phase}$; Figure 6c shows the Fourier residual structure of Figure 3b with loss function $L_{1}+0.2L_{1phase}$; Figure 6d shows the Fourier residual structure of Figure 3b with loss function $L_{1}+0.5L_{1phase}$; Figure 6e shows the Fourier residual structure of Figure 3b with loss function $L_{1}+0.05L_{2phase}$; Figure 6f shows the Fourier residual structure of Figure 3b with loss function $L_{1}+0.2L_{2phase}$; Figure 6g shows the Fourier residual structure of Figure 3b with loss function $L_{1}+0.5L_{2phase}$; Figure 6h shows the Fourier residual structure of Figure 3c with loss function $L_{1}$; Figure 6i shows the Fourier residual structure of Figure 3c with loss function $L_{1}+0.05L_{1phase}$; Figure 6j shows the Fourier residual structure of Figure 3c with loss function $L_{1}+0.2L_{1phase}$; Figure 6k shows the Fourier residual structure of Figure 3c with loss function $L_{1}+0.5L_{1phase}$; Figure 6l shows the Fourier residual structure of Figure 3c with loss function $L_{1}+0.05L_{2phase}$; Figure 6m shows the Fourier residual structure of Figure 3c with loss function $L_{1}+0.2L_{2phase}$; and Figure 6n shows the Fourier residual structure of Figure 3c with loss function $L_{1}+0.5L_{2phase}$.

### 3.4. Comparison with State-of-the-Art Denoising Methods

In this section, we present the results of our network for denoising grayscale and color images corrupted by AWGN and compare them with the results of DnCNN, UNet, and SUNet.

#### 3.4.1. Gray Image Denoising

Table 2 lists the results of image denoising using different denoising models on the BSD68 dataset and displays the parameter quantities and runtime of different models used for image denoising tasks in the last column. On the BSD68 dataset with a noise level of 50, our method improved the PSNR by 1.4323 dB and the SSIM by 0.0208 compared to those of DnCNN. On the BSD68 dataset with a noise level of 50, our method improved the PSNR by 1.0126 dB and the SSIM by 0.0122 compared to those of UNet. Compared to SUNet, a model with a large number of parameters as a transformer, our method improved the PSNR by 0.9677 dB and the SSIM by 0.0033.

Table 3 and Table 4, respectively, list the results of image denoising using different denoising models on the Set12 dataset for images with noise levels of 10, 30, and 50. On the Set12 dataset with noise levels of 10 and 30, our method achieved ideal results. For example, when the noise level was 30, our method improved the average PSNR by 0.3959 dB and the average SSIM by 0.0168 compared to those of DnCNN. Compared with UNet, our method had an average PSNR increase of 1.1033 dB and an average SSIM increase of 0.0116. Compared to SUNet, our method had an average PSNR improvement of 0.9487 dB and an average SSIM improvement of 0.0074. Although our method did not achieve the best average PSNR on the Set12 dataset with a noise level of 50, both the average SSIM and the SSIM of a single image achieved higher results than those obtained by DnCNN, UNet, and SUNet. On balance, our network showed advantages in the noise removal of gray images. Figure 7 shows a comparison of the visual effects of image “test006” in the BSD68 dataset after denoising using different gray-level image denoising models with a noise level of 50. Figure 8 shows a comparison of the visual effects of image “Monarch” in the Set12 dataset after denoising using different grayscale image denoising models with a noise level of 30. Figure 9 shows a comparison of the visual effects of image “Lena” in the Set12 dataset after denoising using different gray-level image denoising models with a noise level of 50.

#### 3.4.2. Color Image Denoising

Table 5 lists the PSNR and SSIM values under different noise levels compared on the CBSD68 and Kodak24 datasets, respectively. The last column displays the parameter quantities and runtime of different methods used for image denoising tasks. On the CBSD68 dataset with a noise level of 50, our method improved the PSNR by 1.1443 dB and the SSIM by 0.0282 compared to those of DnCNN. On the Kodak24 dataset with a noise level of 50, our method improved the PSNR by 1.2443 dB and the SSIM by 0.0339 compared to those of DnCNN. On the CBSD68 dataset with a noise level of 50, our method improved the PSNR by 0.7419 dB and the SSIM by 0.0180 compared to those of UNet. On the Kodak24 dataset with a noise level of 50, our method improved the PSNR by 2.2228 dB and the SSIM by 0.0243 compared to those of Une. Compared with SUNet, our method had a much smaller number of parameters in the network. Although the improvement in the PSNR and SSIM is minimal on the CBSD68 dataset with a noise level of 50, on the Kodak24 dataset with a noise level of 50, our method increased the PSNR by 1.1673 dB and the SSIM by 0.0103. In summary, our network exhibited a clear advantage in the noise removal performance of color images. Figure 10 shows the comparison results of the visual effects of denoising image “14037” in the CBSD68 dataset with a noise level of 30 using different denoising models. Figure 11 shows the comparison results of the visual effects of denoising image “21077” in the CBSD68 dataset with a noise level of 50 using different denoising models. Figure 12 shows the comparison results of the visual effects of denoising image “kodim20” in the Kodak24 dataset with a noise level of 50 using different denoising models.

## 4. Conclusions

In this paper, we proposed a method for image denoising based on Fourier priors. We designed and implemented residual blocks for amplitude spectrum and phase spectrum processing of noisy images. Experiments on synthetic noise datasets showed that our method can effectively recover clean images from noisy images and that the content and details are well preserved, which significantly improves the performance of image denoising. In the future, we will attempt to explore the frequency domain features of noisy real-world images.

## Figures and Tables

**Figure 1 entropy-25-01418-f001:**
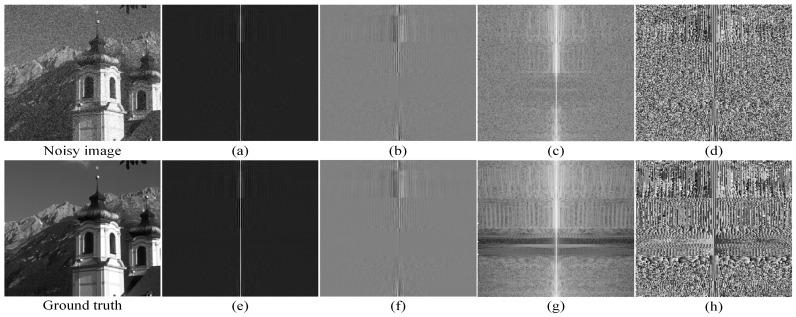
Visualized image after Fourier transform. (**a**) represents the real part diagram of the complex matrix of the noisy image after Fourier transform and frequency domain centralization; (**b**) represents the imaginary part diagram of the complex matrix of the noisy image after Fourier transform and frequency domain centralization; (**c**) represents the amplitude spectrum obtained by the Fourier transform calculation of the noisy image after centralization; (**d**) represents the phase spectrum obtained by the Fourier transform calculation of the noisy image after centralization; (**e**) represents the real part diagram of the complex matrix of the ground truth after Fourier transform and frequency domain centralization; (**f**) represents the imaginary part diagram of the complex matrix of the ground truth after Fourier transform and frequency domain centralization; (**g**) represents the amplitude spectrum obtained by the Fourier transform calculation of the ground truth image after centralization; (**h**) represents the phase spectrum obtained by the Fourier transform calculation of the ground truth image after centralization.

**Figure 2 entropy-25-01418-f002:**
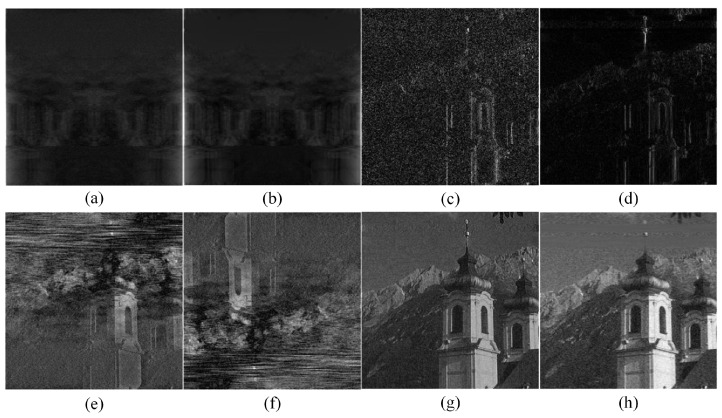
(**a**) represents the image reconstructed using only the phase spectrum from its noise image; (**b**) represents the image reconstructed using only the phase spectrum from its ground truth; (**c**) represents the image reconstructed using only the amplitude spectrum from its noise image; (**d**) represents the image reconstructed using only the amplitude spectrum from its ground truth; (**e**) represents the image reconstructed by combining the amplitude spectrum of the noise image rotated 180° and the phase spectrum of the original noise image; (**f**) represents the image reconstructed by combining the phase spectrum of the noise image rotated 180° and the amplitude spectrum of the original noise image; (**g**) represents the image reconstructed by combining the amplitude spectrum of the noisy image and the phase spectrum of the ground truth; (**h**) represents the image reconstructed by combining the amplitude spectrum of the ground truth and the phase spectrum of the noisy image.

**Figure 3 entropy-25-01418-f003:**
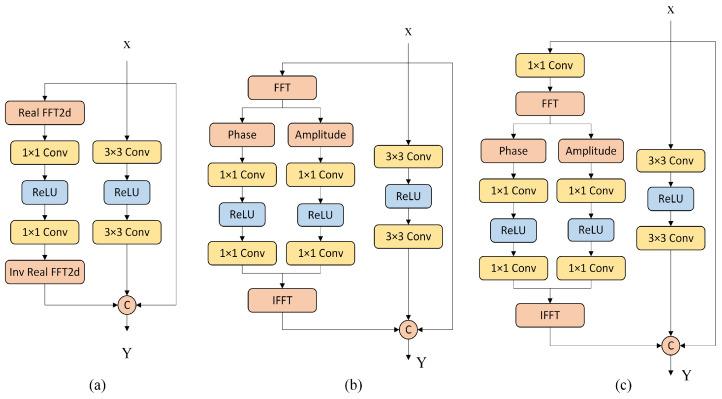
(**a**) represents the existing Res FFT-Conv Block; (**b**,**c**) represent the proposed improved Res FFT-Conv Block, where (**b**) represents our proposed RFAPB.

**Figure 4 entropy-25-01418-f004:**
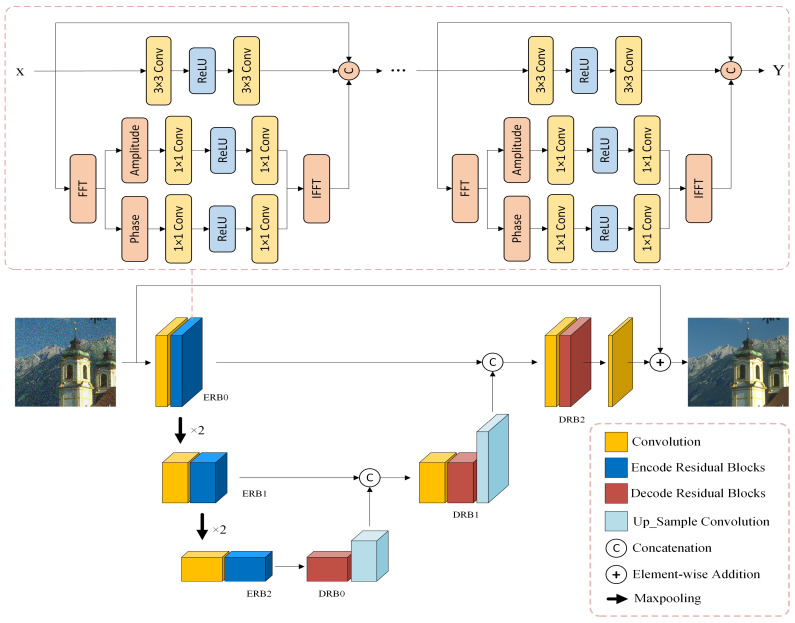
Network structure. Our network structure embeds the RFAPB module within a three-layer UNet architecture. The abbreviations ERB and DRB stand for encoding residual blocks and decoding residual blocks, respectively, as illustrated in the example.

**Figure 5 entropy-25-01418-f005:**
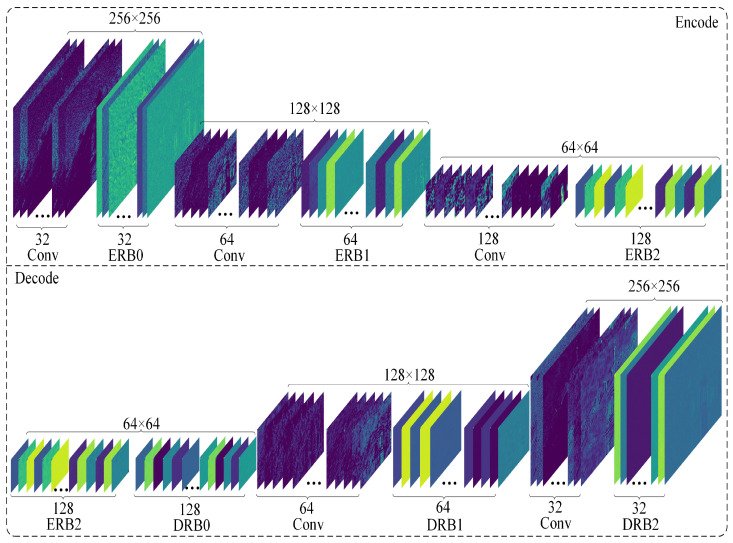
The intermediate results at different stages.

**Figure 6 entropy-25-01418-f006:**
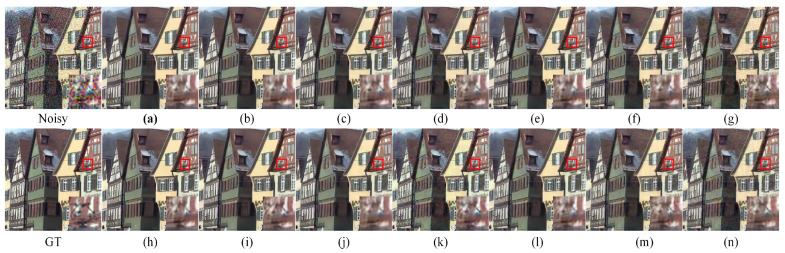
A comparison of the visual effects of denoising color image “kodim08” from the Kodak24 dataset using different Fourier transform structures for denoising is performed. The image is corrupted with additive Gaussian white noise σ=50. The bottom right corner of each subfigure shows the enlarged result within the red box. The first row represents the visualization results of the noisy image and the Fourier residual structure of Figure 4b with different loss functions. The second row represents the visualization results of the ground truth and the Fourier residual structure of Figure 4c with different loss functions. The best results are highlighted in bold. Noisy image: 18.2552 dB/0.5348; ground truth: *∞*/1.0; (**a**): 30.5806 dB/0.8451; (**b**): 30.4021 dB/0.8381; (**c**) 30.3749 dB/0.8351; (**d**) 24.3112 dB/0.8286; (**e**): 24.1747 dB/0.8269; (**f**): 23.7677 dB/0.8061; (**g**): 22.5183 dB/0.7448; (**h**): 30.2183 dB/0.8328; (**i**): 30.4640 dB/0.8391; (**j**): 24.5194 dB/0.8362; (**k**): 22.8274 dB/0.7684; (**l**): 24.1793 dB/0.8246; (**m**): 29.7990 dB/0.8095; (**n**): 22.8878 dB/0.7675.

**Figure 7 entropy-25-01418-f007:**
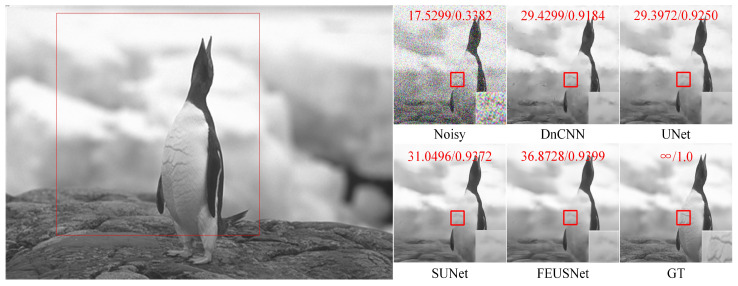
A comparison of the visual effects of the image “test006” in the BSD68 dataset after denoising using different grayscale image denoising models with a noise level of 50 is presented. The PSNR and SSIM values are calculated based on the patch in the upper part of the subgraph and highlighted in red font.

**Figure 8 entropy-25-01418-f008:**
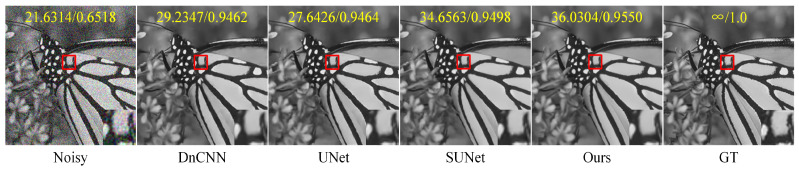
A comparison of the visual effects of the image “Monarch” in the Set12 dataset after denoising using different grayscale image denoising models is presented. The image is adversely affected by additive Gaussian white noise with a standard deviation of σ=30. The PSNR and SSIM values are displayed in the upper part of the image and highlighted in yellow font. Additionally, an enlarged result image is shown within the red box in the bottom right corner of the subgraph.

**Figure 9 entropy-25-01418-f009:**
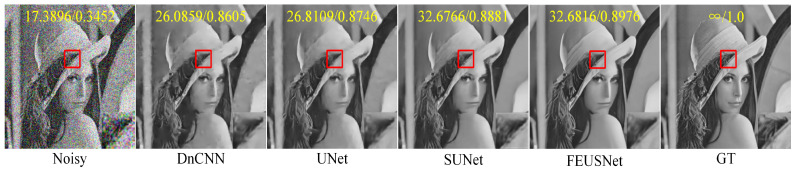
A comparison of the visual effects of the image “Lena” in the Set12 dataset after denoising using different grayscale image denoising models is presented. The image is affected by additive Gaussian white noise with a standard deviation of σ=50. The PSNR and SSIM values are displayed in the upper part of the image and highlighted in yellow font. Additionally, an enlarged result image is shown within the red box in the bottom right corner of the subgraph.

**Figure 10 entropy-25-01418-f010:**
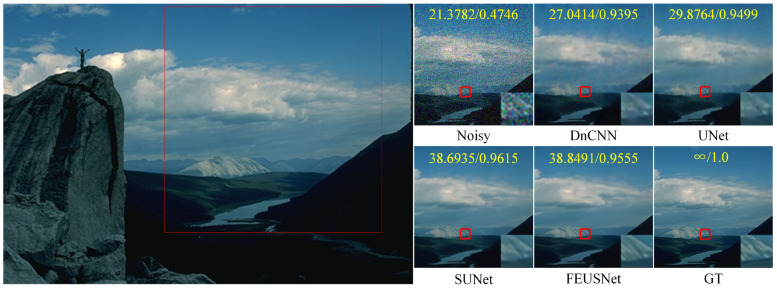
A comparison of the visual effects of the image “14037” in the CBSD68 dataset after denoising using different color image denoising models is presented. The image is affected by additive Gaussian white noise with a standard deviation of σ=30. The PSNR and SSIM values are calculated based on the patch in the upper part of the subgraph and displayed in yellow font. Furthermore, the enlarged result within the red box of the subgraph is shown in the bottom right corner.

**Figure 11 entropy-25-01418-f011:**
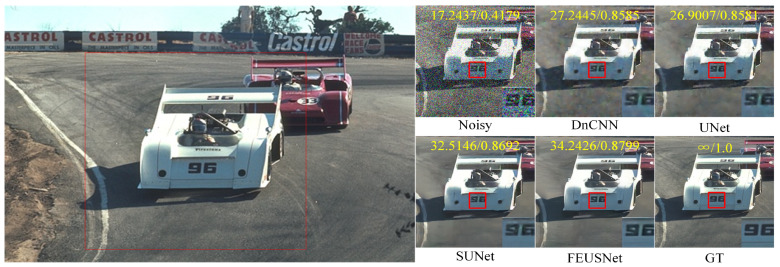
A comparison of the visual effects of the image “21077” in the CBSD68 dataset after denoising using different color image denoising models is presented. The image is affected by additive Gaussian white noise with a standard deviation of σ=50. The PSNR and SSIM values are calculated based on the patch in the upper part of the subgraph and displayed in yellow font. Furthermore, the bottom right corner of the subgraph shows the enlarged result within the red box.

**Figure 12 entropy-25-01418-f012:**
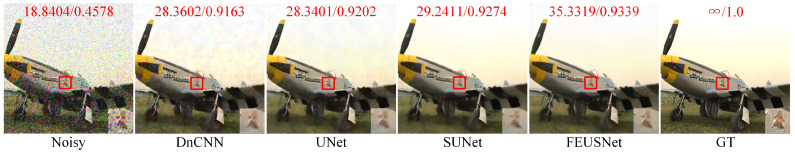
A comparison of the visual effects after denoising the image “kodim20” in the Kodak24 dataset using different color image denoising models is presented. The image is affected by additive Gaussian white noise with a standard deviation of σ=50 The PSNR and SSIM values are displayed in the upper part of the image, highlighted in red font. Additionally, the enlarged result image is shown within the red box located at the bottom right corner of the subgraph.

**Table 1 entropy-25-01418-t001:** Verify the results of residual structure image denoising using different Fourier transforms and highlight the best results in bold font.

Methods	Loss	λ=0		λ=0.05		λ=0.2		λ=0.5	
		**PSNR**	**SSIM**	**PSNR**	**SSIM**	**PSNR**	**SSIM**	**PSNR**	**SSIM**
(b)	$L_{1}+\lambda L_{1phase}$	**33.5579**	**0.8872**	33.2095	0.8836	32.7649	0.8751	31.8952	0.8579
	$L_{1}+\lambda L_{2phase}$			32.3945	0.8666	31.6962	0.8453	29.1296	0.7655
(c)	$L_{1}+\lambda L_{1phase}$	33.3783	0.8843	33.2428	0.8837	32.8874	0.8770	29.4967	0.7906
	$L_{1}+\lambda L_{2phase}$			32.5373	0.8695	31.6969	0.8454	30.1490	0.7995

**Table 2 entropy-25-01418-t002:** The results of image denoising using different denoising models on the BSD68 dataset show that all PSNR and SSIM values are averaged across the entire dataset, with the best results highlighted in bold font.

Methods	BSD68						Parms	Runtime
	σ=10		σ=30		σ=50			
	**PSNR**	**SSIM**	**PSNR**	**SSIM**	**PSNR**	**SSIM**		
DnCNN [22]	33.6233	0.9552	28.9922	0.8806	26.4376	0.8269	558K	0.005 s
UNet [37]	33.9969	0.9640	29.7034	0.8931	26.8573	0.8355	34M	0.009 s
SUNet [32]	35.2309	0.9671	29.5802	0.8964	26.9022	0.8444	99M	0.048 s
FEUSNet	**35.8763**	**0.9689**	**30.2040**	**0.9004**	**27.8699**	**0.8477**	8M	0.044 s

**Table 3 entropy-25-01418-t003:** On the Set12 dataset, for images with noise levels of 10, 30, and 50, the PSNR (in dB) of grayscale images denoised using different denoising models is obtained, and the best result is highlighted in bold font.

Images	C.man	House	Peppers	Starfish	Monarch	Airplane	Parrot	Lena	Barbara	Boat	Man	Couple	Average
Noise level	10												
DnCNN [22]	31.8587	34.9391	27.0164	32.6365	40.3418	**35.9615**	33.1813	30.9360	34.5168	33.0507	34.0274	31.6149	33.3401
UNet [37]	33.2413	35.6834	26.4148	32.0360	40.1507	32.9350	34.9532	38.1159	31.1866	33.2963	26.5380	32.7682	33.1100
SUNet [32]	**33.7737**	**41.9341**	32.3558	39.1856	33.9902	30.5177	35.5054	**39.7368**	**36.9011**	**40.0563**	27.6437	33.1838	35.3987
FEUSNet	33.5682	41.8812	**36.5702**	**40.7204**	**40.6232**	32.0950	**35.5087**	35.5617	33.4224	35.1235	**34.0752**	**34.6913**	**36.1534**
Noise level	30												
DnCNN [22]	29.0558	36.6891	25.7730	28.8950	29.2347	25.4943	29.6536	**33.5781**	**32.6968**	28.4647	**30.2848**	28.2205	29.8367
UNet [37]	29.3609	36.7196	23.5930	28.5871	27.6426	24.3376	29.9903	28.3835	26.7738	**34.8503**	30.2311	29.0818	29.1293
SUNet [32]	29.6134	36.7858	25.6224	28.3282	34.6563	23.9163	30.1507	27.8893	26.9235	34.7974	24.7896	27.9343	29.2839
FEUSNet	**29.9776**	**38.1233**	**25.7778**	**34.9665**	**36.0304**	**27.1347**	**30.4399**	29.2414	28.5770	29.2499	23.5881	**29.6847**	**30.2326**
Noise level	50												
DnCNN [22]	27.0796	27.8077	**33.0292**	26.0415	26.3214	23.6338	27.4224	26.0859	**24.6739**	32.0285	28.3184	26.1363	27.3816
UNet [37]	27.3405	34.2780	28.0408	26.2381	**33.1297**	24.1050	27.6724	26.8109	24.2559	26.3609	22.0707	26.5218	27.2354
SUNet [32]	27.5922	**35.0753**	26.9206	**32.3959**	27.1645	24.0077	27.7915	32.6766	23.7805	**32.5406**	22.3230	26.3605	**28.2191**
FEUSNet	**27.8703**	27.6870	25.1088	26.9962	26.7924	**24.8094**	**27.9388**	**32.6816**	22.9595	26.5815	**28.9749**	**27.1161**	27.1264

**Table 4 entropy-25-01418-t004:** On the Set12 dataset, for images with noise levels of 10, 30, and 50, the results of denoising using different denoising models for grayscale images are SSIM, and the best result is highlighted in bold font.

Images	C.man	House	Peppers	Starfish	Monarch	Airplane	Parrot	Lena	Barbara	Boat	Man	Couple	Average
Noise level	10												
DnCNN [22]	0.9257	0.9605	0.9638	0.9650	0.9800	0.9587	0.9666	0.9608	0.8904	0.9635	0.8930	0.9621	0.9492
UNet [37]	0.9377	0.9646	0.9645	0.9631	0.9813	0.9633	0.9751	0.9632	0.9376	0.9634	0.8948	0.9673	0.9563
SUNet [32]	**0.9455**	0.9672	0.9695	0.9688	0.9811	0.9678	0.9774	0.9676	0.9550	0.9684	0.9009	0.9693	0.9615
FEUSNet	0.9452	**0.9728**	**0.9742**	**0.9753**	**0.9841**	**0.9692**	**0.9793**	**0.9726**	**0.9720**	**0.9733**	**0.9123**	**0.9738**	**0.9670**
Noise level	30												
DnCNN [22]	0.8851	0.9313	0.9318	0.9249	0.9462	0.9105	0.9288	0.9094	0.8463	0.8966	0.7942	0.8990	0.9003
UNet [37]	0.8912	0.9330	0.9319	**0.9262**	0.9464	0.9144	0.9345	0.9190	0.8483	0.9083	0.8067	0.9057	0.9055
SUNet [32]	0.9016	0.9368	0.9362	0.9209	0.9498	0.9165	0.9369	0.9217	0.8594	0.9095	**0.8156**	0.9115	0.9097
FEUSNet	**0.9030**	**0.9399**	**0.9413**	0.9259	**0.9550**	**0.9213**	**0.9384**	**0.9307**	**0.9070**	**0.9152**	0.8083	**0.9193**	**0.9171**
Noise level	50												
DnCNN [22]	0.8449	0.9062	0.8980	0.8756	0.9094	0.8767	0.8971	0.8605	0.7972	0.8489	0.7299	0.8413	0.8571
UNet [37]	0.8540	0.9135	0.9061	0.8808	0.9158	0.8805	0.9037	0.8746	0.8082	0.8581	0.7336	0.8485	0.8648
SUNet [32]	0.8663	0.9224	0.9112	0.8865	0.9218	0.8865	0.9080	0.8881	0.8199	0.8667	0.7405	0.8566	0.8729
FEUSNet	**0.8679**	**0.9235**	**0.9145**	**0.8952**	**0.9239**	**0.8869**	**0.9099**	**0.8976**	**0.8405**	**0.8720**	**0.7560**	**0.8691**	**0.8798**

**Table 5 entropy-25-01418-t005:** The results of image denoising using different denoising models on the CBSD68 and Kodak24 datasets show that all PSNR and SSIM values are averaged across the entire dataset, with the best results highlighted in bold font.

Methods	CBSD68						Kodak24						Parms	Runtime
	σ=10		σ=30		σ=50		σ=10		σ=30		σ=50			
	**PSNR**	**SSIM**	**PSNR**	**SSIM**	**PSNR**	**SSIM**	**PSNR**	**SSIM**	**PSNR**	**SSIM**	**PSNR**	**SSIM**		
DnCNN [22]	33.5717	0.9618	28.7173	0.8922	26.6094	0.8306	33.1976	0.9560	29.4191	0.8889	28.2674	0.8332	558K	0.007 s
UNet [37]	34.8622	0.9664	29.4497	0.8987	27.0964	0.8408	35.3172	0.9618	29.9454	0.8944	27.2889	0.8428	34M	0.011 s
SUNet [32]	35.0486	0.9706	29.8743	0.9081	**27.8385**	0.8542	35.0168	0.9663	30.1521	0.9038	28.3444	0.8568	99M	0.059 s
FEUSNet	**36.6497**	**0.9732**	**30.6664**	**0.9119**	27.8383	**0.8588**	**37.1193**	**0.9716**	**31.3508**	**0.9123**	**29.5117**	**0.8671**	8M	0.051 s

## Data Availability

The data presented in this study are available on request from the corresponding author. Data are not publicly available due to privacy considerations.

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
