# Peer review of "FEUSNet: Fourier Embedded U-Shaped Network for Image Denoising"

_entropy, 2023, doi:10.3390/e25101418_

Round 1
Reviewer 1 Report
In this paper, a new deep learning model for image de-noising is proposed.
The introduced model involves Fourier Transform towards
improving the denoising capabilities of the model.
After studying the manuscript and the related references the following comments are stated:
1) The contribution of this work is clearly defined.
2) The technical presentation is adequate and in depth.
3) The experimental results justify the superiority of the proposed model
over the other methods in the literature.
4) There is no doubt that this is a high quality work.
Author Response
Thank you very much for your positive and valuable comments on our work.
Reviewer 2 Report
It is well known that the Fourier method provides the best filtering for white Gaussian noise. For more complex noise cases, wavelet filtering methods compete, and recently Shearlet methods have shown excellent results. The application of today's popular neural network technologies of artificial intelligence is significantly complicated by the necessity of initial marking of the initial data and large expenditure of computer time. In the peer-reviewed paper an attempt is made to combine the advantages of two filtering methods - Fourier and deep learning. A large number of computational experiments related to hyperparameter enumeration are performed. The superiority of the new method in comparison with previously known methods of using neural networks for processing and filtering photo images is obtained. Unfortunately, the paper does not present working materials of the experiments, for example, intermediate results of filtering at different stages of numerical image processing. As a result, the contribution of the Fourier algorithm and the deep learning algorithm remains unclear. Doubts arise about the necessity of additional processing of the Fourier results to improve the final result. Without filling this research gap, this paper cannot be recommended for publication.
With the advent of the ChatGPT service, the quality of English papers submitted for review has improved significantly.
Reviewer 3 Report
Majors:
mean SSIM or PSNR are not informative, please show distributions also
Minors:
Eq.5 and more - Please change R to Re and I to Im
l.129 fast Fourier transform <- Fast Fourier Transform or simply FFT
section 3.2.1 information about framework, etc. desired
eq. 11 \cross <- \cdot or empty
eq. 12 \cross <- \cdot or empty
Reviewer 4 Report
I believe the manuscript presents good piece of research but I would really like authors to improve the manuscript based on my comments before publication,
@ Are there any potential biases or limitations in the automated image denoising approach that should be addressed?
@ Are there any unexpected findings or patterns in the data that should be explored further?
@ A clear rationale for why this work is important and timely should be provided in the introduction.
@ In Figure 3 and figure 4 don't present the best performing values by red color. Kindly change it to bold.
@ As this manuscript is about image denoising by NN I believe there must be a small discussion (if testing not possible) about image quality assessment. Include such recent articles and discuss about advancements about image quality assessments techniques,
- https://doi.org/10.1016/j.displa.2021.102101
- https://doi.org/10.1016/j.cviu.2023.103695
- 10.3389/fnins.2023.1222815
- https://doi.org/10.1007/s11042-022-13040-6
Round 2
Reviewer 2 Report
1) "2. Fourier Embeded U-Shaped Network We first introduce the Fourier prior in Section 2.1, where we conjecture and prove the characteristics of amplitude and phase spectra in noisy images."
- I suggest that you add the phrase "the well-known characteristics" and provide a couple of classic books on the Fourier transform.
The optimal procedure for high frequency Fourier filtering is to remove frequency harmonics above some threshold. For complex cases the problem of "soft threshold" arises. The "deep learning" method proposed below may be effective in solving this problem.
2) "A widely used residual Fast Fourier Transform module based on ReLU is only utilized to concatenate the real and imaginary parts in the last dimension after the Fourier transform." I'm sorry, I don't understand if this refers to threshold filtering of a signal? Would you be kind enough to give the analytic form of the ReLU transform!
3) Finally, the analytical form of formula (8) is far from being so obvious, compared to formula (3).
4) "In Figure 5, we present the intermediate results at different stages. The entire network is divided into two layers (Encode and Decode), and we visualize all the high-dimensional feature maps for each intermediate filter." Sorry I don't see any intermediate result as compared to Figure 4!
Reviewer 3 Report
ok
Author Response
Thank you very much for your advice.